# Power output from 12 brands of contemporary LED light-curing units measured using 2 brands of radiometers

**Cristiane Maucoski[1], Richard B. Price[2]\*, Cesar A. Arrais[1], Braden Sullivan[2]**

**1** Department of Restorative Dentistry, State University of Ponta Grossa, Ponta Grossa, Parana, Brazil,
**2** Department of Dental Clinical Sciences, Dalhousie University, Halifax, Nova Scotia, Canada

\* rbprice@dal.ca

**Data Availability Statement:** All relevant data are within the paper and its Supporting Information files.

**Funding:** The authors received no specific funding for this work. However, CM received a travel grant

## Abstract

### Background

Given the increasing use of photo-activated resins in dentistry, dentists and researchers need a user-friendly dental radiometer to measure the power output from dental light-curing units (LCUs).

### Objective

Our goal was to measure the accuracy of two brands of dental radiometers in reporting the power (mW) from twelve brands of contemporary LCUs compared to a 'gold standard' (GS) reference value obtained from an integrating sphere attached to a fiberoptic spectroradiometer.

### Methods

The power output was measured from two units of 12 brands of LCUs, five times on the "GS" system, five times on two Bluephase Meter II dental radiometers, and five times on two Mini Gig hand-held spectroradiometers. The emission spectrum was also recorded using the 'GS' integrating sphere. The power values reported by each meter were subjected to t-tests to compare the two examples of each LCU, and 3-way ANOVA followed by Bonferroni's post-hoc tests. Regression analyses were also performed to determine the relationship between the data from the hand-held radiometers and the 'GS' integrating sphere.

### Results

There was a large difference in the power values (mW) and the emission spectra from the 12 brands of LCUs on their standard-settings (p<0.001). Except for one LCU (Dental Spark @ 15.1%), the differences between the two LCUs of the same brand were less than 5.3% when measured using the 'GS' integrating sphere. Regression analyses showed a highly significant agreement between the power values reported from the two brands of radiometers and the 'GS' integrating sphere (R$^2$ > 98%).

in the form of a MITACS award [IT26826] and an internal research grant from the Faculty of Dentistry, Dalhousie University. CM's visit to Canada was also supported by the Coordination for the Improvement of Higher Education Personnel in the form of a grant [CAPES, grant no. 88881.622852/2021-01]. These funders had no role in the study design, data collection and analysis, decision to publish, or preparation of the manuscript.

**Competing interests:** The authors have read the journal's policy and have the following competing interests: Ivoclar Vivadent and Ultradent loaned the authors the Bluephase Meter II and the Mini Gig radiometers used in this study. There are no patents, products in development or marketed products associated with this research to declare. This does not alter our adherence to PLOS ONE policies on sharing data and materials.

## Conclusion

We concluded that the power values reported from both brands of dental radiometers we tested were accurate, provided that the light source did not emit wavelengths of light that were beyond the radiometer's detection limit.

## 1. Introduction

After diabetes and cardiovascular diseases, the treatment of oral disease now accounts for the third-highest expenditure among non-communicable diseases in European Union [1]. Most teeth are now restored with light-cured dental resins [2] because of the Minamata Agreement to phase down the use of dental amalgam [3]. In 2012, it was estimated that more than 260 million direct resin-based composite (RBC) dental restorations were placed annually [4]. However, much of the treatment that dentists provide is the replacement of previous restorations [5], and this is mainly done using light-cured adhesives and light-cured resins. If these light-cured resins are not sufficiently photocured, they will not reach their intended properties [6, 7], and the economic impact of millions of inadequately photopolymerized RBC restorations failing even just a few years prematurely is both enormous and unacceptable. However, this could be the current situation because every study published has reported that most dentists do not know if the light-curing units (LCUs) they are using deliver an acceptable light output [8–12]. Furthermore, most dentists do not own or use a dental radiometer [8–12], probably because, unlike radiographic or sterilization equipment, dentists are not required to monitor the LCU they use in their dental office, and even low power LCUs will photocure the top surface of the resin. Consequently, many offices will purchase 'budget' LCUs [11], not realizing that even though the irradiance values at the light tip may be comparable, they may deliver much less power than more expensive LCUs from major manufacturers. This misconception may explain why a Cochrane review published in 2021 reported that direct RBC restorations placed in posterior teeth have almost double the failure rate of amalgam restorations [13]. Several other publications have also reported high failure rates for RBCs [5, 14] and that the median longevity of posterior RBCs placed in dental offices is less than 7 years [15, 16].

Research on the properties of dental resins, the effects of different photo-activation protocols, and on factors affecting the adhesion between the resin, tooth and restorative materials has become prolific [17]. Unfortunately, very few dental researchers report accurate information about the light that their resin specimens received [17]. Most contemporary LCUs used by clinicians and researchers use a light-emitting diode (LED) as the light source [9, 17–19]. These LEDs offer several advantages over other light sources. They are solid-state, robust, lightweight, and since they are efficient at producing light, the LCU can be battery operated [20, 21]. In addition, LEDs can provide a longer working life than filament or spark-based light sources [21]. However, the nature of the LED emitters used in dental LCUs means that, unlike quartz tungsten-halogen (QTH) lights, LED lights only produce a narrow band of wavelengths of light [20–25]. To deliver a broader spectrum of light, some LED curing lights now include two or more kinds of LED emitters, each producing a different band of wavelengths [21, 23, 24, 26, 27]. Unfortunately, the delivery of various discrete wavelength bands from these LCUs may lead to incompatibilities between the LCU and the dental radiometer, thus producing inaccurate results. For example, in one study, the differences between the 'gold standard' method of measuring the irradiance and the irradiance values from the dental radiometer ranged from as little as 7% to as much as 535% [28], depending on the LCU tested. In

addition, some meters have only a narrow aperture to the light sensor inside the radiometer [25, 28, 29]. If the LCU emits a non-homogenous light beam profile across the light tip, then depending on the position of the light tip over this small aperture into the radiometer, the irradiance value may be high or low [25, 28–31]. Thus, it is not surprising that several previous studies have reported large differences between the irradiance values provided by the manufacturer, those obtained from dental radiometers, those measured by dentists, and those derived from laboratory-grade equipment [25, 28–31].

Most clinicians will choose their LCU based on the price and radiant exitance (tip irradiance) value in the belief that a light that delivers a high irradiance value is better and more powerful than one that delivers a lower value. This may not be correct. To determine the radiant exitance from the LCU, the ISO 10650 standard [32] requires that the radiant flux (power in Watts) be measured. This power value is then divided by the tip area of the LCU to provide a single averaged radiant exitance (tip irradiance) value (in mW/cm$^2$), but the radiant flux (power) value that is used to calculate the radiant exitance is rarely reported. However, since the radiant exitance (tip irradiance) is a calculated value, even small changes in the tip diameter will have a large effect on the irradiance [11, 23]. In fact, two LCUs can produce the same tip irradiance, yet due to differences in the area of their light tips, their power outputs can easily be two or three times different. In addition, most researchers fail to adequately report the energy (radiant exposure) and what wavelengths of light were delivered to the specimens [17]. This problem was highlighted by a review of how the light irradiation conditions had been described in 300 laboratory studies published between January 2017 and May 2018 [17]. An irradiance value was reported in only 231 (77%) out of the 300 articles. The irradiance was measured in 50 articles (16.7%) using a dental radiometer, 11 articles (3.7%) used a power meter, only 15 articles also reported the emission spectrum from the LCU, and 132 articles did not state how the irradiance was measured. This failure to adequately describe the light received by the resin specimens in almost every publication between January 2017 and May 2018 may lead to inappropriate treatment recommendations.

Therefore, both dental clinicians and researchers need to easily and accurately measure the output from the LCU. Although hand-held dental radiometers are readily available, most are inaccurate [25, 28–31], they only report an irradiance value, and few dentists use them [8, 9, 12, 18, 33]. Recently, two hand-held radiometers that use new technologies and claim improved accuracy have become available. The Bluephase Meter II (Ivoclar Vivadent, Amherst, NY, USA) dental radiometer has a large aperture and can measure the output from light tips that are up to 13 mm in diameter. This radiometer reports the power (mW) from the light, and when the light tip diameter is entered into the meter software, it also reports the irradiance (mW/cm$^2$). This dental radiometer costs less than $450, and the manufacturer claims that it can measure the radiant power output from 380–550 nm with an accuracy of ±10% [34]. Although this dental radiometer has received favorable reports, it still may have difficulty accurately reporting the output from some LED units [28, 31], and it does not report the emission spectrum from the LCU. Alternatively, the Mini Gig (Ultradent, South Jordan, UT, USA) radiometer has a small integrating sphere attached to a Gigahertz Optik MSC15 spectroradiometer (Gigahertz Optik, Türkenfeld, Germany). The Mini Gig can be computer controlled, and the results are easily exported. Although this radiometer costs approximately $2,000, this is a laboratory-grade spectroradiometer that can provide the accurate spectral radiant power recordings according to the ISO/IEC 17025 standard 'General requirements for the competence of testing and calibration laboratories' [35, 36] that researchers require when describing the light output from the LCU used in their studies.

Thus, since the radiant flux (power) measurement forms the foundation of the ISO 10650 standard [32] for reporting the radiant exitance from dental LCUs, and this power value is

unaffected by the tip area or light beam homogeneity [23], the power and not the irradiance from the LCU should be a better and more reliable value to report when comparing radiometers.

Therefore, in this study, we aimed to:

1. Report the power (mW) from twelve contemporary LED LCUs using two brands of radiometers and compare these values to 'gold standard' reference values obtained from a laboratory-grade integrating sphere attached to a fiberoptic spectrometer.

2. Report the emission spectra from these twelve LED LCUs.

3. Determine if both brands of radiometers could provide precise and accurate power values compared to the 'gold standard' system.

Our research hypotheses were:

1. There would be no difference ($p \geq 0.05$) in the power values and the emission spectra from the 12 brands of contemporary LCUs.

2. When measured on the 'gold standard' integrating sphere system, the power values from two examples of each brand of LCU tested would be within ± 10% of each other.

3. The two examples of each brand of radiometer would report power values that were within ± 10% of each other.

4. The power values reported by the two brands of radiometer tested would be accurate and precise compared to the power values obtained from the 'gold standard' system.

## 2. Materials and methods

### 2.1 Sample selection

We measured 7 brands of single peak and 5 brands of multiple peak wavelength LED LCUs (Fig 1). All the LCUs had been used for several hours in other laboratory studies, and they represented a wide range of contemporary LCUs that dentists and researchers currently use. The brand, serial number, manufacturer, output mode, tip diameter, and type of LCU (single emission peak or multiple emission peak LCU) are reported in Table 1. Four of these LCUs were purchased from online sellers on the Internet and we considered them to be 'budget' LCUs. The other eight LCUs were from six major dental manufacturers. Two lower-priced and two higher-priced hand-held radiometers (Fig 2 and Table 2) that could both report the power in mW were tested.

### 2.2 Light output measurements

To ensure that a representative example of each brand was measured, the light outputs from two units of each brand of LCU were measured in their standard output mode for 10 seconds. These LCUs were not new, but they were in good condition and showed no visible signs of damage (Fig 1). The power output from each LCU was measured five times using two new units of the Bluephase Meter II and five times using two examples of the Mini Gig spectroradiometer (Fig 2 and Table 2). The LCUs were recharged after every 5 exposures to ensure that their batteries were always adequately charged. The LCUs and meters were used in random order and at room temperature (20˚C±1).

The power values obtained from each radiometer were compared to a 'GS' reference value obtained from a laboratory-grade spectroradiometer attached to an integrating sphere. Using

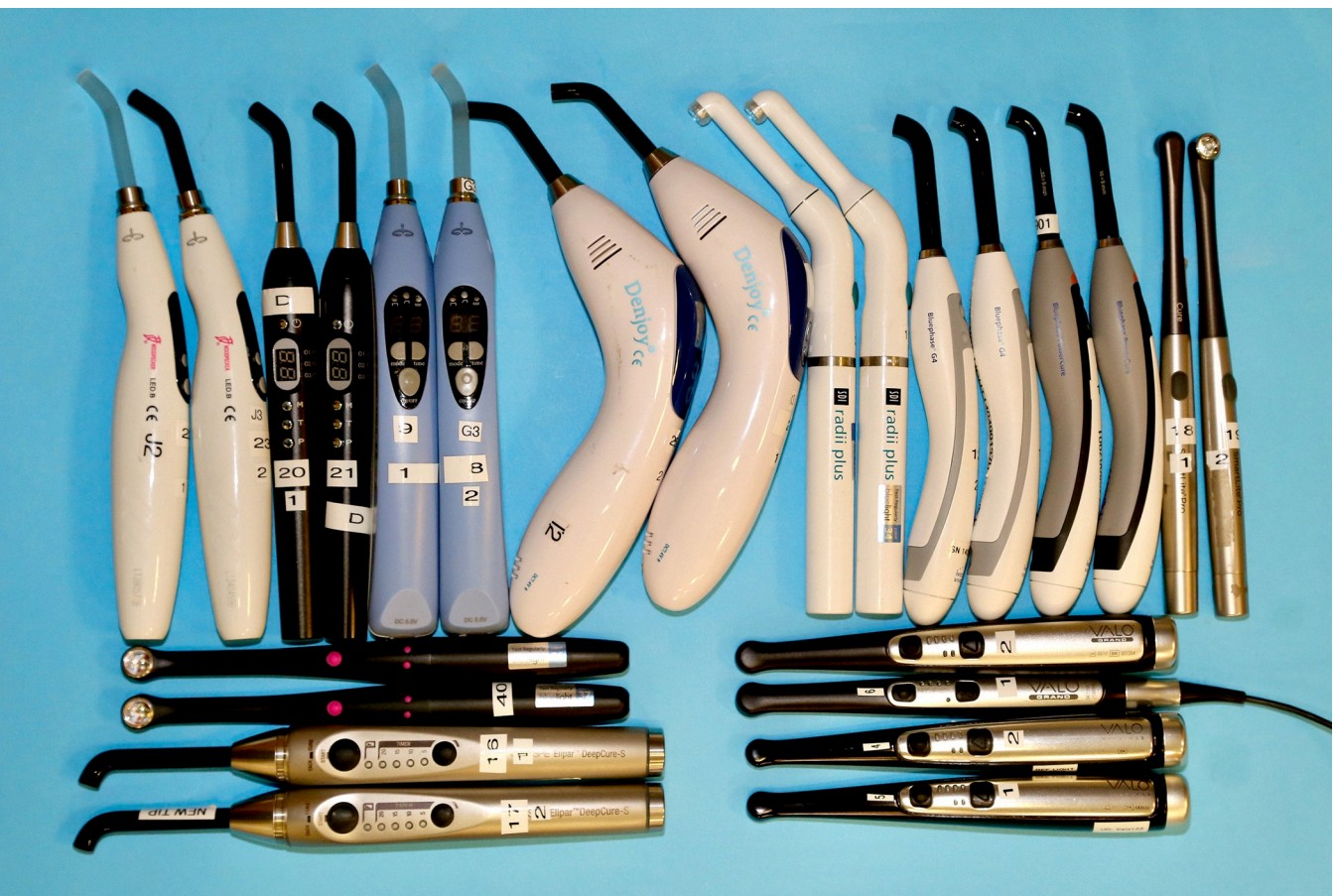

**Fig 1. Units #1 and #2 of each brand of light-curing unit used in the study.**

previously described methods [28, 37], the tip of each LCU was placed at the 16-mm diameter entrance into the 6″ integrating sphere (Labsphere, North Sutton, NH, USA) that was coupled to a fiber optic Flame T spectrometer (Ocean Insight, Orlando, FL, USA). This 16-mm diameter aperture was large enough to capture all of the light from the LCUs. An internal traceable light source, SCL 600 (Labsphere) within the sphere, was used to calibrate the system before beginning the measurements [28, 37]. Five measurements were made of each LCU. In addition, the GS recorded the emission spectra from each LCU.

## 2.3 Statistical analysis

The power values of the tested LCUs were subjected to normality and homoscedasticity tests using Shapiro-Wilk and Levene tests, respectively. After they had passed these tests, the data were subjected to a 3-way ANOVA followed by Bonferroni's post hoc test using the SPSS v20 statistical program (SPSS Inc, IBM Company, Armonk, NY, USA) and the results are reported in Tables 3 and 4. Finally, regression analyses (Origin Pro, Northampton, MA, USA) were used to evaluate the relationship between power values from the hand-held radiometers and those from the 'GS' integrating sphere. For a dental LCU, the Pinkwave LCU has an unusually broad emission spectrum that ranged from 390 to 870 nm and the Bluephase Meter II is not designed to measure light outside of the 380 to 550 nm range. Therefore, since the power

**Table 1. Light curing units (LCUs) and information provided by the manufacturers.**

| Light Curing Unit | Code | Manufacturer | Serial Number | Single Peak/Multi-Peak Emission | Tip diameter (mm) | Wavelength range (nm) | Irradiance (mW/cm$^2$) Tolerance (±%) |
|---|---|---|---|---|---|---|---|
| SmartLite Pro #1 | a#1 | Dentsply Sirona, Charlotte, NC, USA | H00045 | **Single Peak** | 10 | 450–480 | 1,200 |
| SmartLite Pro #2 | a#2 | | H00466 | | | | |
| DeepCure #1 | b#1 | 3M Oral Care, St. Paul, MN, USA | 939112012777 | **Single Peak** | 10 | 430–480 | 1,470 (10%/+20%) |
| DeepCure #2 | b#2 | | 933112003463 | | | | |
| Dental Spark #1 | c#1 | Foshan Keyuan Medical Equipment Foshan City, Guangdong, China | SK13L0201324 | **Single Peak** | 8 | 430–485 | 1,400 |
| Dental Spark #2 | c#2 | | SK13L0301315 | | | | |
| Denjoy #1 | d#1 | Denjoy Dental Co, Changsha, China | DYD21302089 | **Single Peak** | 8 | 450–470 | 1,000 1,400 |
| Denjoy #2 | d#2 | | DYD21302064 | | | | |
| Woodpecker LED.D #1 | e#1 | Guilin Woodpecker Medical Instrument Co., Guilin, Guangxi, China | D12020417A | **Single Peak** | 8 | 420–480 | 1,000–1,200 |
| Woodpecker LED.D #1 | e#1 | | D12020417A | | | | |
| Woodpecker LED.B #1 | f#1 | | L12B0572B | **Single Peak** | 8 | 420–480 | 850–1,000 |
| Woodpecker LED.B #2 | f#2 | | L1340459B | | | | |
| SDI radii plus #1 | g#1 | SDI, Bayswater Victoria, Australia | Nothing visible | **Single Peak** | 8 | 440–480 | 1,500 |
| SDI radii plus #2 | g#2 | | Nothing visible | | | | |
| Valo Grand #1 | h#1 | Ultradent, South Jordan, UT, USA | T10172 | **Multi-Peak** | 12 | 385–515 | 900 (±10%) |
| Valo Grand #2 | h#2 | | S01264 | | | | |
| Bluephase G4 #1 | i#1 | Ivoclar Vivadent, Schaan, Liechtenstein | 1404001370 | **Multi-Peak** | 10 | 385–515 | 1,200 (±10%) |
| Bluephase G4 #2 | i#2 | | 1400002115 | | | | |
| Valo Cordless #1 | j#1 | Ultradent, South Jordan, UT, USA | C43122 | **Multi-Peak** | 10 | 385–515 | 900 (±10%) |
| Valo Cordless #2 | j#2 | | C11296 | | | | |
| Bluephase PowerCure #1 | k#1 | Ivoclar Vivadent, Schaan, Liechtenstein | 1428005297 | **Multi-Peak** | 9 | 385–515 | 1,200 (±10%) |
| Bluephase PowerCure #2 | k#2 | | 1428007901 | | | | |
| PinkWave #1 | l#1 | Vista Dental Products, Racine, WI, USA | 00107C | **Multi-Peak** | 12 | 395–900 | > 1,515 |
| PinkWave #2 | l#2 | | 00225C | | | | |

values reported for the Pinkwave using the Bluephase Meter II were obviously incorrect, the data from this LCU was removed from the regression analyses.

## 3. Results

### 3.1 Power and emission spectra

Representative real-time radiant power outputs and emission spectra of the 12 brands of LCU measured using the 'GS' integrating sphere system over the 10-s exposures are reported in Figs 3 and 4, respectively. Fig 3 shows that the power output from the 12 LCUs was stable (flat) when the LCUs were turned on. Of note, in Fig 3, the Pinkwave LCU would remain on for 20-s and the Radii Plus for 60-s, but only the output in the first 15-s is reported.

Fig 4 shows that 7 (a to g) of the LCUs (SmartLite Pro, DeepCure S, Dental Spark, Denjoy, Woodpecker LED B and D, and Radii Plus) were single emission peak LCUs. The remaining 5 LCUs (h to l) were broader spectrum multi-emission peak LCUs. The Pinkwave LCU (l) emitted 4 distinct wavelength peaks ($\lambda 1 = 411$ nm, $\lambda 2 = 472$ nm, $\lambda 3 = 633$ nm, and $\lambda 4 = 857$ nm)

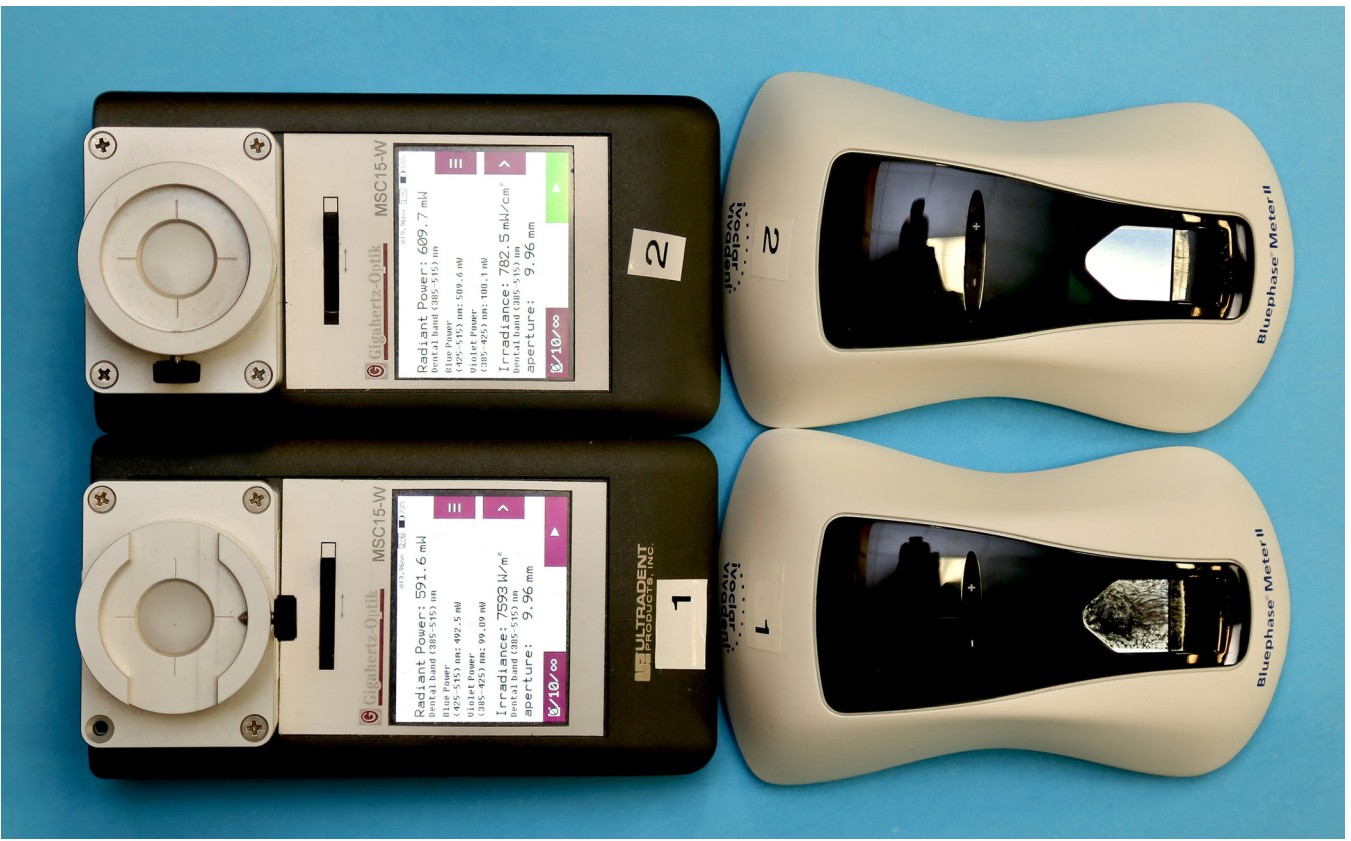

**Fig 2. The two examples of the Bluephase Meter II and the Mini Gig radiometer used in the study.**

and was noticeably different from the other LCUs. Fig 5 illustrates the power values from the 24 LCUs when measured with the 'GS' and using the other 4 radiometers. Apart from the Pinkwave LCU when it was measured using both examples of the Bluephase Meter II, Fig 5 shows that all the measurement methods produced similar power values from the LCUs. However, statistically, there were significant differences in the power values reported from the various radiometers when measuring each LCU (p<0.05). Fig 6 compares the 12 brands of LCUs as a percentage of the maximum power (929 mW) that was measured using the 'GS' system. This maximum value was measured from the SmartLite Pro unit, but the Valo Grand and Pinkwave both delivered similar power values (918 and 905 mW, respectively). However, 5 LCUs (Dental Spark, Denjoy, Woodpecker LED.D, Woodpecker LED.B, and the SDI Radii Plus) delivered only 59%, 50%, 41%, 39% and 33% respectively of the power from the SmartLite Pro.

**Table 2. Information provided by the manufacturers about their radiometers.**

| Radiometer | Serial Number | Manufacturer | Spectral Range (nm) | Sensor Window Aperture (mm) | Claimed Accuracy |
|---|---|---|---|---|---|
| Bluephase Meter II #1 | 1300011617 | Ivoclar Vivadent, Schaan, Liechtenstein | 380–550 | 15 | ±10% |
| Bluephase Meter II #2 | 1300011631 | | | | |
| Mini Gig #1 | 45047 | Gigahertz Optik, Türkenfeld, Germany | 360–830 | 15 | ±4% |
| Mini Gig # 2 | 34845 | | | | |

**Table 3. Mean power (mW) ± standard deviation (SD) and coefficient of variation (CV) values recorded using the 'gold standard' or using the two examples (#1 and #2) of each brand of hand-held radiometer when measuring unit #1 of each brand of LCU.**

| LCU #1 | Gold Standard | | | | Bluephase #1 | | | | Bluephase #2 | | | | Mini Gig #1 | | | | Mini Gig #2 | | | |
|---|---|---|---|---|---|---|---|---|---|---|---|---|---|---|---|---|---|---|---|---|
| | Mean Power (mW) | SD (mW) | CV | Sig. Equivalent Groups | Mean Power (mW) | SD (mW) | CV | Sig. Equivalent Groups | Mean Power (mW) | SD (mW) | CV | Sig. Equivalent Groups | Mean Power (mW) | SD (mW) | CV | Sig. Equivalent Groups | Mean Power (mW) | SD (mW) | CV | Sig. Equivalent Groups |
| **Valo Grand** | 918 | 8.8 | 1.0 | Ac | 903 | 5.9 | 0.7 | Bc | 953 | 6.0 | 0.6 | Bb | 946 | 5.2 | 0.6 | Ab | 994 | 5.2 | 0.5 | Aa |
| **PinkWave** | 913 | 2.9 | 0.3 | Ac | 1389 | 13.7 | 1.0 | Ab | 1427 | 14.9 | 1.1 | Aa | 914 | 16.0 | 1.8 | Bc | 915 | 25.5 | 2.8 | Cc |
| **SmartLite Pro** | 896 | 13.8 | 1.5 | Ac | 813 | 4.2 | 0.5 | Ce | 841 | 1.9 | 0.2 | Cd | 937 | 10.3 | 1.1 | Ab | 967 | 7.0 | 0.7 | Ba |
| **DeepCure** | 754 | 2.9 | 0.4 | Bb | 727 | 4.5 | 0.6 | Dc | 756 | 4.9 | 0.7 | Dab | 731 | 1.4 | 0.2 | Cc | 770 | 2.4 | 0.3 | Da |
| **Bluephase G4** | 715 | 10.5 | 1.5 | Cb | 679 | 2.2 | 0.3 | Ec | 719 | 3.7 | 0.5 | Eb | 709 | 3.8 | 0.5 | Db | 735 | 4.6 | 0.6 | Ea |
| **Valo Cordless** | 661 | 11.7 | 1.8 | Bc | 638 | 7.8 | 1.2 | Fd | 665 | 7.8 | 1.2 | Fc | 687 | 2.5 | 0.4 | Eb | 713 | 2.8 | 0.4 | Fa |
| **PowerCure** | 581 | 8.4 | 1.4 | Eab | 528 | 5.1 | 1.0 | Gd | 556 | 3.7 | 0.7 | Gc | 568 | 3.2 | 0.6 | Fbc | 592 | 2.3 | 0.4 | Ga |
| **Dental Spark** | 466 | 30.4 | 6.5 | Fa | 384 | 3.5 | 0.9 | Ic | 399 | 5.6 | 1.4 | Ic | 425 | 2.5 | 0.6 | Hb | 439 | 6.2 | 1.4 | Ib |
| **Denjoy** | 465 | 6.8 | 1.5 | Fab | 417 | 6.0 | 1.5 | Hc | 432 | 12.3 | 2.8 | Hc | 463 | 3.3 | 0.7 | Gb | 480 | 2.4 | 0.5 | Ha |
| **Woodpecker LED.D** | 382 | 3.7 | 1.0 | Ga | 353 | 6.2 | 1.8 | Jb | 361 | 8.3 | 2.3 | Jb | 379 | 2.5 | 0.7 | Ia | 390 | 3.6 | 0.9 | Ja |
| **Woodpecker LED.B** | 359 | 5.6 | 1.6 | Hab | 321 | 4.8 | 1.5 | Kd | 337 | 4.8 | 1.4 | Kcd | 347 | 6.1 | 1.8 | Jbc | 364 | 3.5 | 1.0 | Ka |
| **SDR Radii Plus** | 288 | 31.9 | 11.1 | Ia | 251 | 6.1 | 2.4 | Lc | 255 | 7.2 | 2.8 | Lc | 271 | 9.4 | 3.5 | Kb | 272 | 13.1 | 4.8 | Lb |

(N = 5 repeated measurements using each meter).

Mean power (mW) values followed by the same letters (upper case letters within the column; lower case letters within the row) are not significantly different (3-way ANOVA p≥0.05)

**Table 4. Mean power (mW) ± standard deviation (SD) and coefficient of variation (CV) values recorded using the 'gold standard' or using the two examples (#1 and #2) of each brand of hand-held radiometer when measuring unit #2 of each brand of LCU.**

| LCU #2 | Gold Standard | | | | Bluephase #1 | | | | Bluephase #2 | | | | Mini Gig #1 | | | | Mini Gig #2 | | | |
|---|---|---|---|---|---|---|---|---|---|---|---|---|---|---|---|---|---|---|---|---|
| | Mean Power (mW) | SD (mW) | CV | Sig. Equivalent Groups | Mean Power (mW) | SD (mW) | CV (mW) | Sig. Equivalent Groups | Mean Power (mW) | SD (mW) | CV | Sig. Equivalent Groups | Mean Power (mW) | SD (mW) | CV | Sig. Equivalent Groups | Mean Power (mW) | SD (mW) | CV | Sig. Equivalent Groups |
| **SmartLite Pro** | 929 | 12.6 | 1.4 | Ac | 851 | 4.4 | 0.5 | Ce | 876 | 7.6 | 0.9 | Cd | 974 | 24.0 | 2.5 | Ab | 1005 | 12.3 | 1.2 | Aa |
| **PinkWave** | 905 | 8.6 | 1.0 | Bd | 1362 | 11.3 | 0.8 | Ab | 1420 | 9.2 | 0.7 | Aa | 889 | 14.9 | 1.7 | Ce | 921 | 13.7 | 1.5 | Ccd |
| **Valo Grand** | 882 | 1.7 | 0.2 | Cc | 886 | 4.1 | 0.5 | Bc | 924 | 6.0 | 0.7 | Bb | 929 | 2.7 | 0.3 | Bb | 978 | 1.8 | 0.2 | Ba |
| **DeepCure** | 765 | 3.3 | 0.4 | Dab | 753 | 1.6 | 0.2 | Dbc | 777 | 5.5 | 0.7 | Da | 742 | 0.6 | 0.1 | Dc | 778 | 5.3 | 0.7 | Da |
| **Bluephase G4** | 709 | 1.9 | 0.3 | Eb | 655 | 9.9 | 1.5 | Ec | 701 | 8.1 | 1.2 | Eb | 710 | 2.5 | 0.4 | Eb | 735 | 4.2 | 0.6 | Ea |
| **Valo Cordless** | 698 | 1.4 | 0.2 | Eb | 661 | 3.7 | 0.6 | Ec | 692 | 8.9 | 1.3 | Eb | 698 | 3.6 | 0.5 | Eb | 720 | 3.7 | 0.5 | Ea |
| **PowerCure** | 601 | 10.9 | 1.8 | Fab | 554 | 4.1 | 0.7 | Fd | 586 | 3.4 | 0.6 | Fbc | 580 | 1.2 | 0.2 | Fc | 603 | 2.1 | 0.4 | Fa |
| **Dental Spark** | 549 | 25.0 | 4.6 | Ga | 464 | 3.6 | 0.8 | Gc | 495 | 6.0 | 1.2 | Gb | 468 | 2.4 | 0.5 | Gc | 485 | 5.6 | 1.2 | Gb |
| **Denjoy** | 459 | 10.1 | 2.2 | Ha | 411 | 6.6 | 1.6 | Hc | 415 | 8.9 | 2.1 | Hc | 440 | 3.4 | 0.8 | Hb | 460 | 1.9 | 0.4 | Ha |
| **Woodpecker LED.D** | 372 | 11.0 | 3.0 | Ia | 329 | 7.4 | 2.2 | Ib | 341 | 4.4 | 1.3 | Ib | 364 | 1.8 | 0.5 | Ia | 378 | 1.6 | 0.4 | Ia |
| **Woodpecker LED.B** | 344 | 3.1 | 0.9 | Ja | 298 | 5.5 | 1.9 | Jb | 313 | 2.9 | 0.9 | Jb | 332 | 3.5 | 1.1 | Ja | 345 | 3.9 | 1.1 | Ja |
| **SDR Radii Plus** | 304 | 18.8 | 6.2 | Ka | 257 | 5.0 | 1.9 | Kc | 255 | 7.9 | 3.1 | Kc | 273 | 11.0 | 4.0 | Kb | 276 | 10.2 | 3.7 | Kb |

(N = 5 repeated measurements using each meter)

Mean power (mW) values followed by the same letters (upper case letters within the column; lower case letters within the row) are not significantly different (3-way ANOVA p≥0.05)

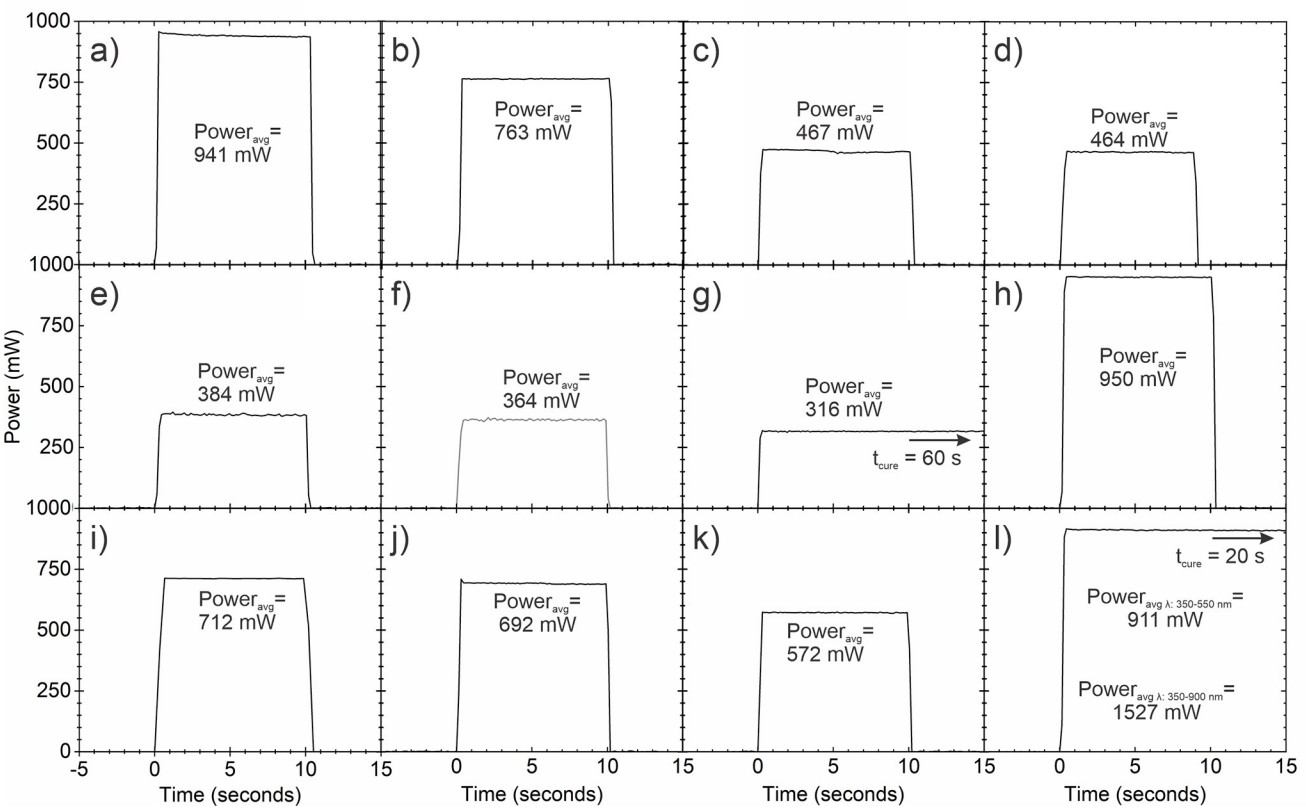

**Fig 3.** Representative 'gold standard' power (mW) emitted by the 12 LCUs evaluated: Single Peak LCUs a) SmartLite Pro; b) DeepCure; c) Dental Spark; d) Denjoy; e) Woodpecker LED D; f) Woodpecker LED B; g) SDI Radii Plus; and Multi-peak LCUs h) Valo Grand; I) Bluephase G4; j) Valo; k) Bluephase Powercure; and l) Pinkwave.

## 3.2 Comparison between radiometers

The mean power values, standard deviation (SD), and coefficient of variation (CV) of LCU units #1 and 2 measured using the 'GS' system and using the four different dental radiometers are reported in Tables 3 and 4. The comparisons between the power values reported by the two examples of each brand of meter showed some significant differences for some LCUs. However, Table 5 shows that the average ± standard deviation % difference in these power values from the two examples of each meter was only 3.8 ± 1.5% for the Bluephase Meter II and 3.4 ± 1.3% for the Mini Gig radiometer.

## 3.3 Comparison between the LCUs

Table 6 shows that the mean power values from unit #1 and #2 of each brand of LCU when measured using the 'GS' were between 0.9% and 5.3% different, with the Dental Spark (C) an outlier at 15.3%. Figs 7 and 8 illustrate the percentage differences of LCU # 1 and #2 on each radiometer compared to the 'GS' power measurement. The highest percentage difference (+51 to 57%) was for the Pinkwave when using the Bluephase Meter II, and these values were obviously incorrect. Otherwise, both meters reported power values that were both slightly above and below the 'GS' value. For the Mini Gig meter #1, the greatest difference from the 'GS' result was -15%. For the Mini Gig meter #2, the greatest difference from the 'GS' power value was -12% (Fig 8). The maximum differences in the power values recorded by the Bluephase

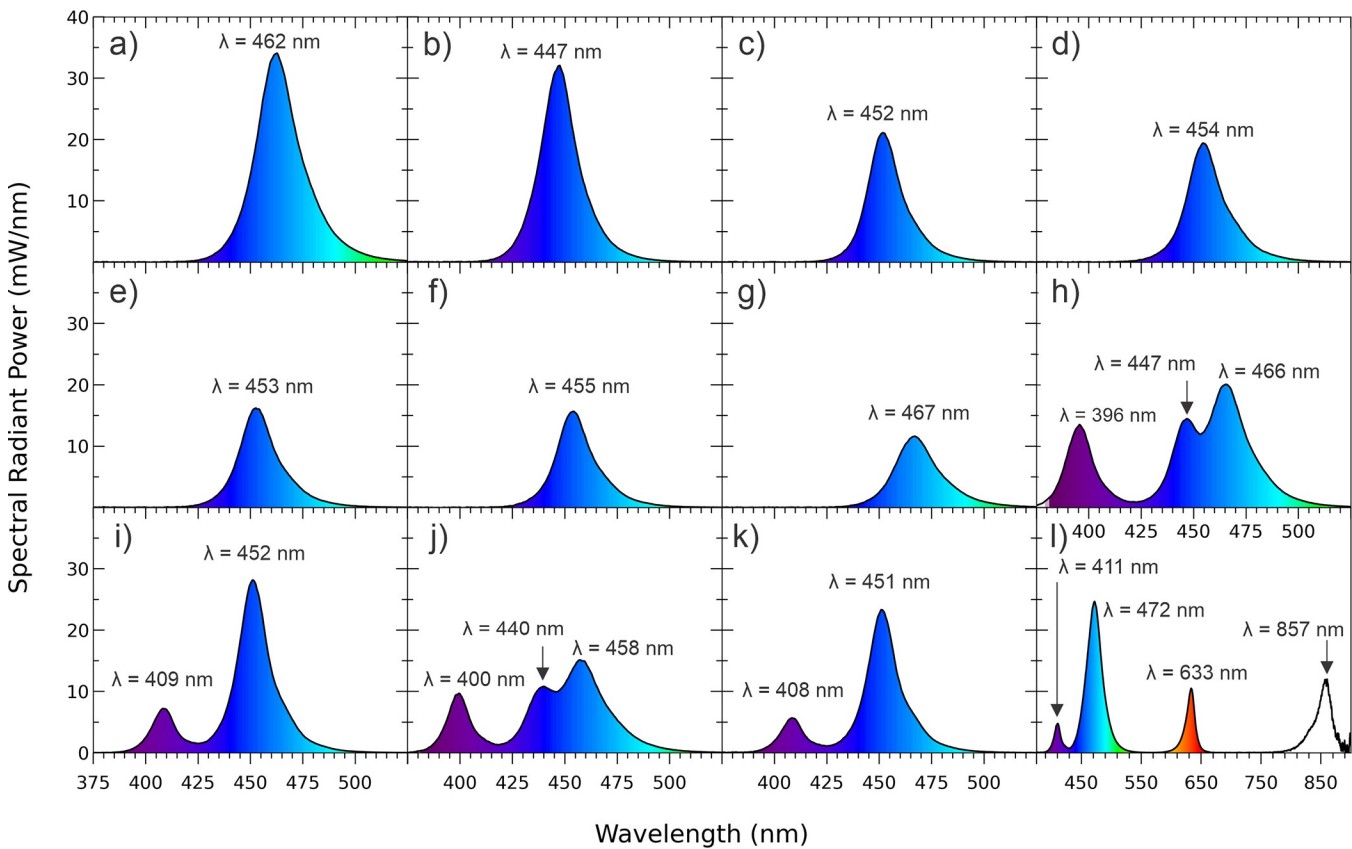

**Fig 4.** Spectral Radiant Powers (mW/nm) emitted by the 12 LCUs with the emission peaks identified: Single Peak LCUs a) SmartLite Pro; b) DeepCure; c) Dental Spark; d) Denjoy; e) Woodpecker LED D; f) Woodpecker LED B; g) SDI Radii Plus; and Multi-peak LCUs h) Valo Grand; I) Bluephase G4; j) Valo; k) Bluephase Powercure; and l) Pinkwave.

Meter II compared to the GS was -18%. All these large % differences were measured from light (C), which we classified as a 'budget LCU', whose standard deviations and coefficient of variation in the 'GS' power values were also large, suggesting that the output from this LCU was unstable.

## 3.4 Comparison between the radiometers and the 'gold standard' (GS) measurement system

The ability of the two brands of meters to report similar power values to those obtained using the 'GS' method within each LCU also varied according to the LCU being measured (Tables 3 and 4 and Figs 7 and 8). When example #1 of each LCU was tested, the power values from three LCUs measured using the Bluephase Meter II were not significantly different from the 'GS' values. In comparison, the power values from six LCUs measured with Mini Gig matched those from the 'GS' integrating sphere. When example #2 of each LCU was measured, the values obtained from five LCUs with Bluephase Meter II were not significantly different ($p>0.05$) from those obtained in the 'GS' system. In comparison, the power values from eight LCUs measured with the Mini Gig radiometer were not significantly different ($p>0.05$) from the 'GS' values. Furthermore, Fig 7 shows that when measuring example #1 of each LCU using the Mini Gig, in 17 out of the 24 power measurements, the difference in the mean power values compared to the 'GS' values was 5% or less, and in all 24 measurements, the difference was less

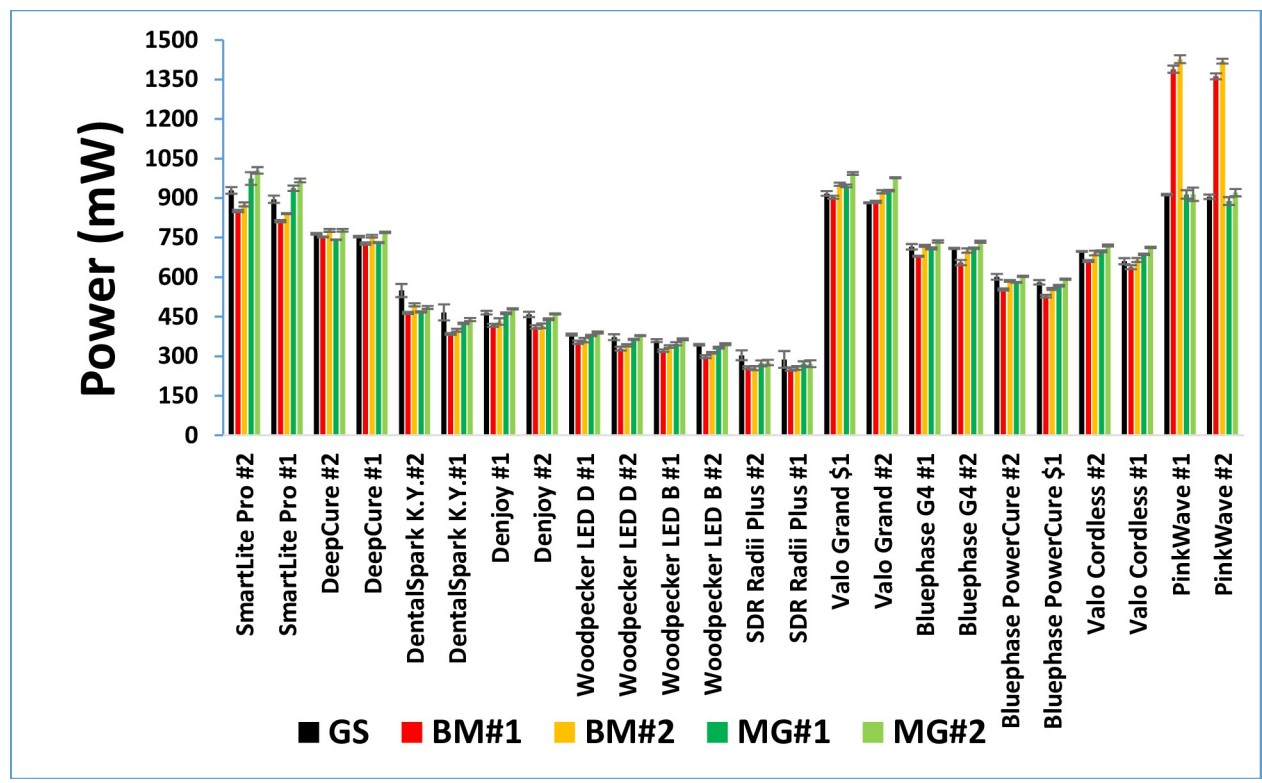

**Fig 5. Mean power output(mW) from the LCUs recorded using the gold standard (GS), and the two examples of the Bluephase Meter II (BM#1 and #2) and the Mini Gig (MG#1 and #2) radiometers.** Note the range in power values from the LCUs and the overall similarity in the radiometer power values for each LCU, apart from the inability of both Bluephase meters to accurately measure the Pinkwave.

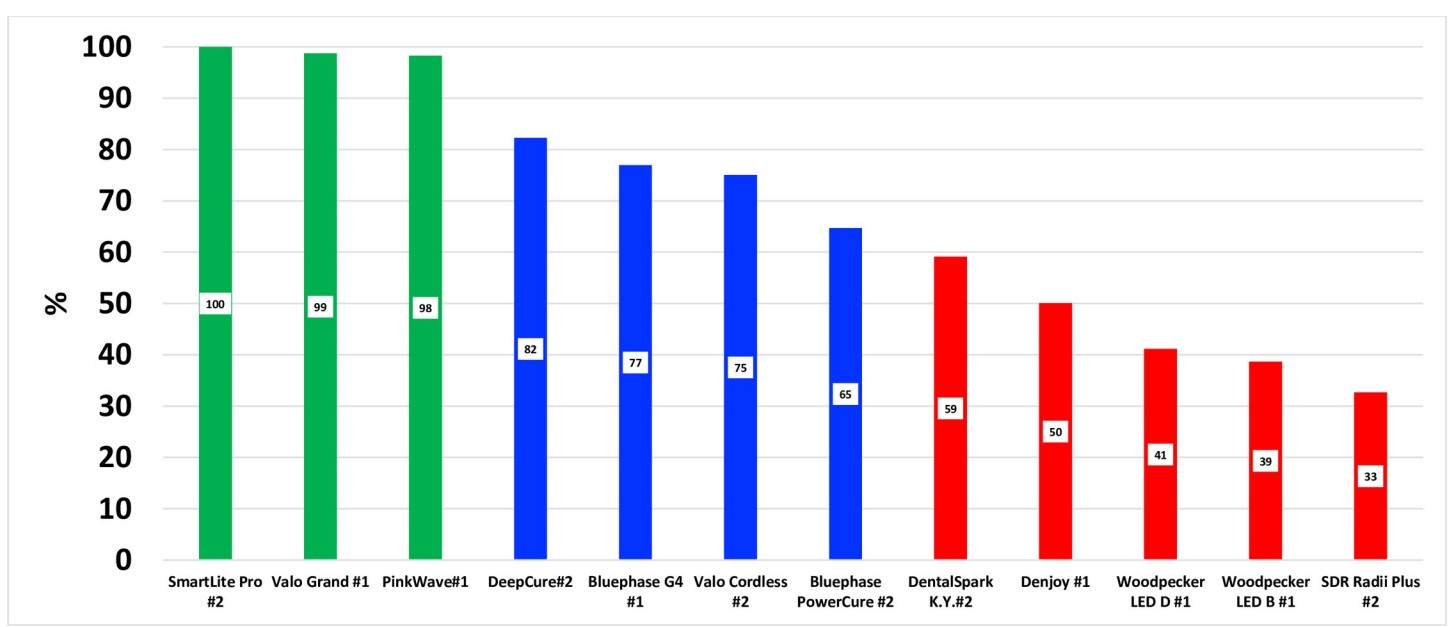

**Fig 6. Comparison of the power output of the 12 LCUs described as a percentage of the most powerful LCU (SmartLite Pro) recorded using the 'gold standard' system.**

**Table 5. Percentage differences between the power values from each LCU (units #1 and #2) recorded using the two Bluephase Meter II and Mini Gig radiometers.**

| LCU and # | Difference between 2 Bluephase II meters | Difference between 2 Mini Gig meters |
|---|---|---|
| | % | % |
| SmartLite Pro #1 | 3.4 | 3.1 |
| SmartLite Pro #2 | 2.9 | 3.1 |
| DeepCure #1 | 3.7 | 5.0 |
| DeepCure #2 | 3.2 | 4.6 |
| DentalSpark K.Y. #1 | 3.6 | 3.3 |
| DentalSpark K.Y. #2 | 6.2 | 3.5 |
| Denjoy #1 | 3.4 | 3.5 |
| Denjoy #2 | 1.0 | 4.3 |
| Woodpecker LED D #1 | 2.2 | 2.9 |
| Woodpecker LED D #2 | 3.7 | 3.8 |
| Woodpecker LED B #1 | 4.7 | 4.6 |
| Woodpecker LED B #2 | 4.7 | 3.8 |
| SDR Radii Plus #1 | 1.5 | 0.1 |
| SDR Radii Plus #2 | 0.8 | 1.2 |
| Valo Grand #1 | 5.2 | 4.8 |
| Valo Grand #2 | 4.1 | 5.0 |
| Bluephase G4 #1 | 5.5 | 3.5 |
| Bluephase G4 #2 | 6.5 | 3.4 |
| Valo Cordless #1 | 4.1 | 3.7 |
| Valo Cordless #2 | 4.4 | 3.1 |
| Bluephase PowerCure #1 | 5.0 | 4.1 |
| Bluephase PowerCure #2 | 5.5 | 3.7 |
| PinkWave #1 | 2.7 | 0.1 |
| PinkWave #2 | 4.0 | 3.5 |
| Mean Difference ±SD | 3.8 ± 1.5 | 3.4 ± 1.3 |

than 10%. Fig 8 shows that when measuring example #2 of each LCU using the Mini Gig, in 18 out of the 24 power measurements, the difference was 5% or less, and in 21 instances, the difference was 10% or less. When using the Bluephase Meter II to measure examples # 1 or 2 of

**Table 6. Percentage difference between the power values from units #1 and #2 of each LCU recorded using the 'gold standard' system.**

| Curing Light | % Difference between unit #1 and unit #2 of each brand of LCU |
|---|---|
| DentalSpark K.Y. | 15.1 |
| Valo Cordless | 5.3 |
| SDR Radii Plus | 5.2 |
| Woodpecker LED B | 4.4 |
| Valo Grand | 3.9 |
| SmartLite Pro | 3.6 |
| Bluephase Power Cure | 3.5 |
| Woodpecker LED D | 2.6 |
| Denjoy | 1.4 |
| Deepcure-S | 1.4 |
| PinkWave | 0.9 |
| Bluephase G4 | 0.9 |

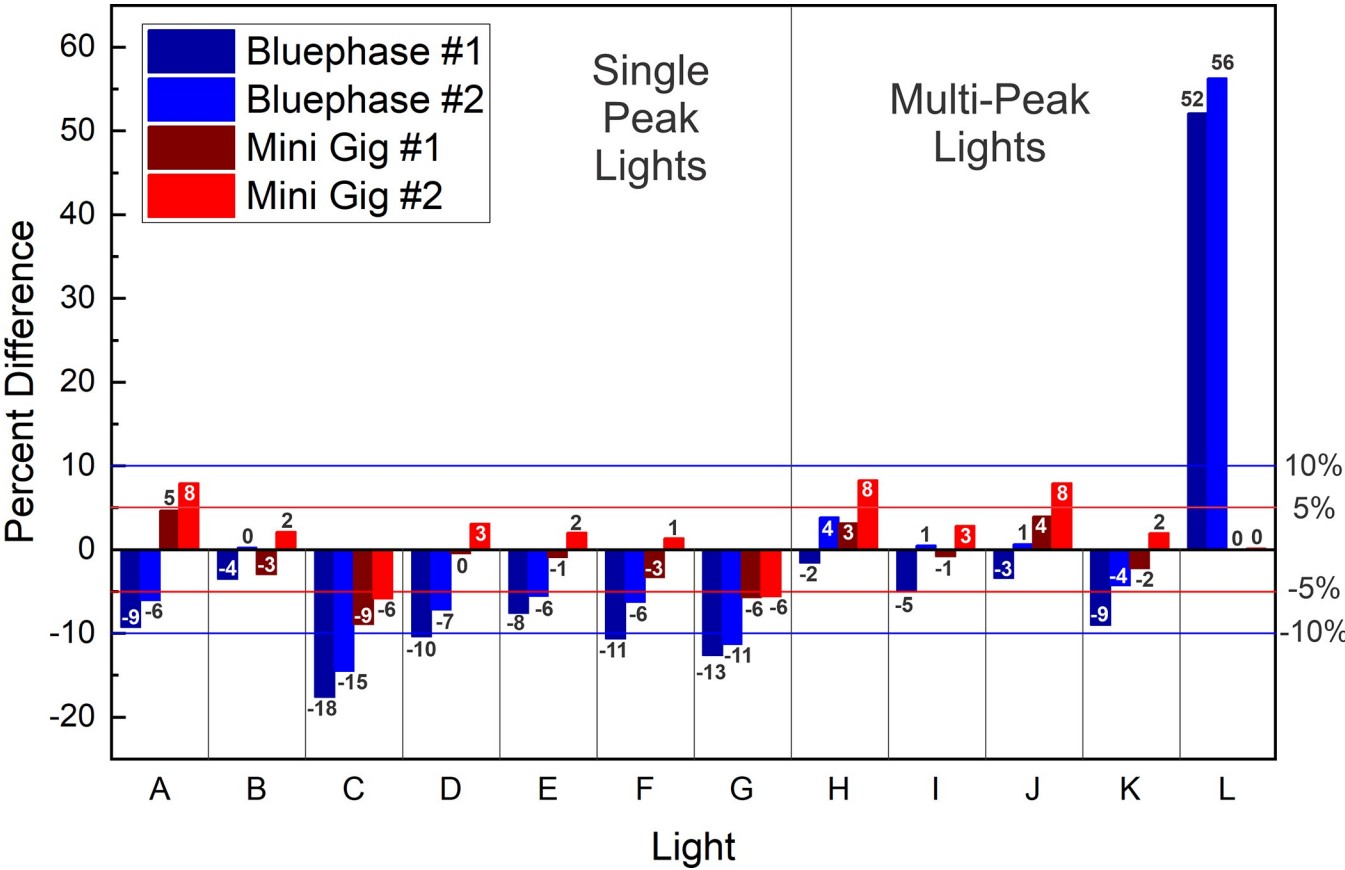

**Fig 7. Percentage difference in the power values between radiometer units #1 and #2 compared to the 'GS' integrating sphere power values measured from unit #1 of each brand of LCU.** Single Peak LCUs A) SmartLite Pro; B) DeepCure; C) Dental Spark; D) Denjoy; E) Woodpecker LED D; F) Woodpecker LED B; G) SDI Radii Plus; and Multi-peak LCUs H) Valo Grand; I) Bluephase G4; J) Valo; K) Bluephase Powercure; and L) Pinkwave.

each brand of LCU, Figs 7 and 8 show that the difference from the 'GS' power value was 10% or less in 17 out of the 24 instances. The power values from the Pinkwave were obviously wrong (51% to 57% greater than the 'GS' value). Figs 7 and 8 both show the power outputs from examples #1 and 2 of the Dental Spark (C), Woodpecker LED.B (F) and SDI Radii Plus (G) LCUs when measured using the Bluephase Meter II were often more than -10% different from the 'GS' power values.

### 3.5 Correlation between the radiometers and the 'GS' measurement system

When the 'GS' power values were compared with all the power values from the two Bluephase Meters II and from the two Mini Gig radiometers (omitting the Pinkwave LCU from the Bluephase Meter II readings only), the regression analyses (Fig 9) showed a highly significant positive relationship between the two brands of meters and the 'GS' measurements ($R^2 > 98\%$).

### 4. Discussion

The purpose of this study was not to measure the light output from brand new curing lights. Instead, we determined if two radiometers could accurately record the power from a representative sample (n = 24) of contemporary LCUs used in dental offices. To provide a broad range of lights, we chose 7 brands of single peak and 5 brands of multiple peak wavelength LCUs.

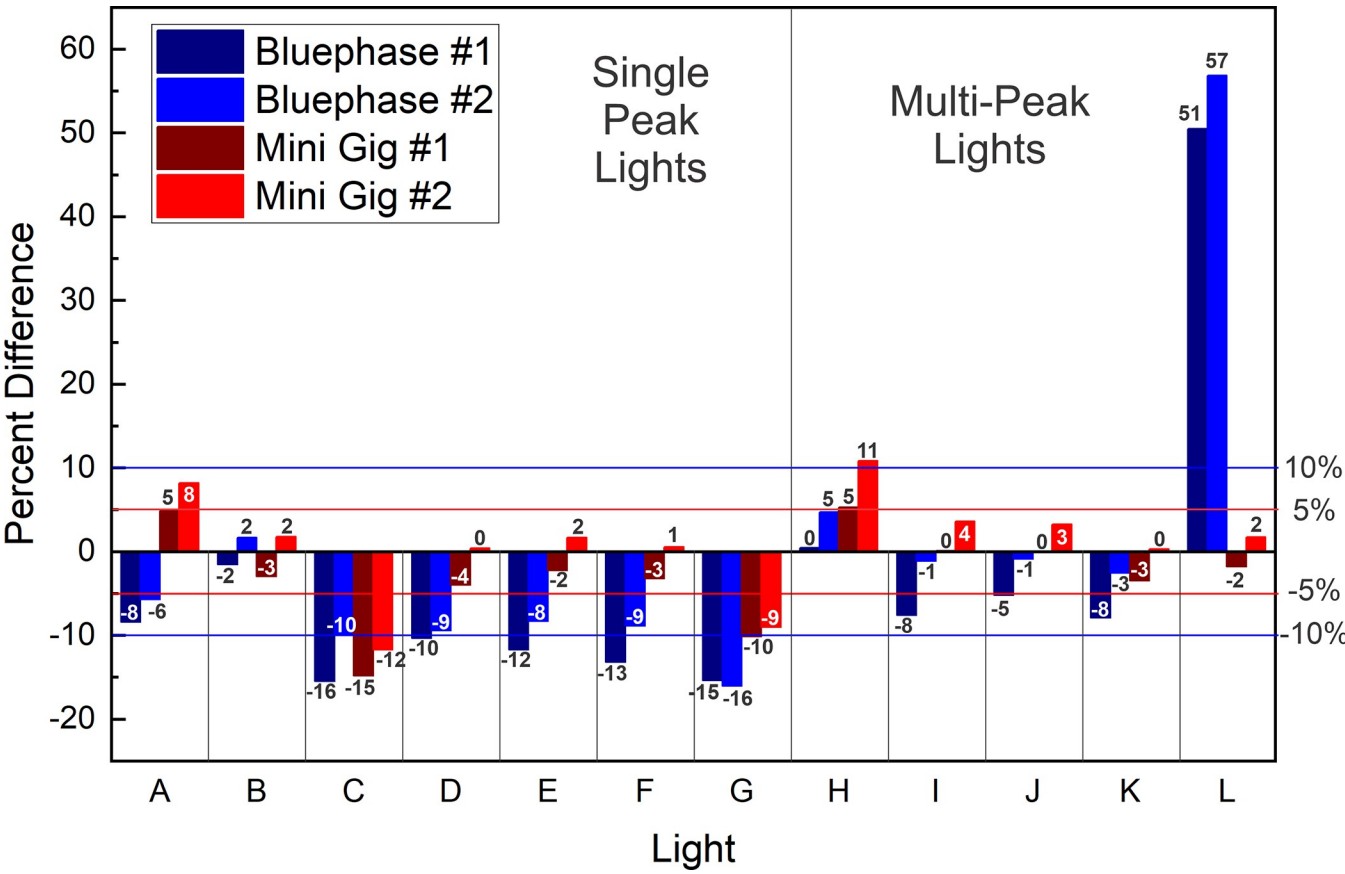

**Fig 8. Percentage difference in the power values between radiometer units #1 and #2 of both radiometers compared to the 'GS' integrating sphere power values measured from unit #2 of each brand of LCU.** Single Peak LCUs A) SmartLite Pro; B) DeepCure; C) Dental Spark; D) Denjoy; E) Woodpecker LED D; F) Woodpecker LED B; G) SDI Radii Plus; and Multi-peak LCUs H) Valo Grand; I) Bluephase G4; J) Valo; K) Bluephase Powercure; and L) Pinkwave.

Since some LCUs had only one output setting, the LCUs were all tested on their standard output settings. We found that the radiant power outputs from the LCUs tested on their standard-setting were markedly different (p<0.001), and the wavelengths of light from these LCUs were also noticeably different (Tables 3 and 4 and Figs 3–6). Five of the LCUs emitted less than 60% of the power emitted by the most powerful LCU, and one LCU emitted less than 33% of the output from the most powerful LCU tested (Fig 6). Thus, the first hypothesis that there would be no difference in the power outputs from the 12 contemporary LCUs that we tested was rejected. In addition, we noted that both meters reported power values that were slightly above and slightly below the 'GS' value (Figs 7 and 8). This supports that the 'GS' power values were accurate, and the measurement differences were based on valid 'GS' power values.

Table 6 reports that, except for the Dental Spark LCU (where the difference was 15.1%), when measured on the 'gold standard system', the differences in the mean power values from the two examples of each brand of LCU were less than 5.3%. Of note, there was less than a 0.9% difference between the two examples of the PinkWave and the two examples of the Bluephase G4 units. Thus, the second hypothesis that the power outputs from the two examples of each LCU brand measured on their standard output settings using the 'GS' system would be within ±10% was accepted for 11 out of the 12 brands of LCU. However, we rejected this hypothesis for the Dental Spark LCU.

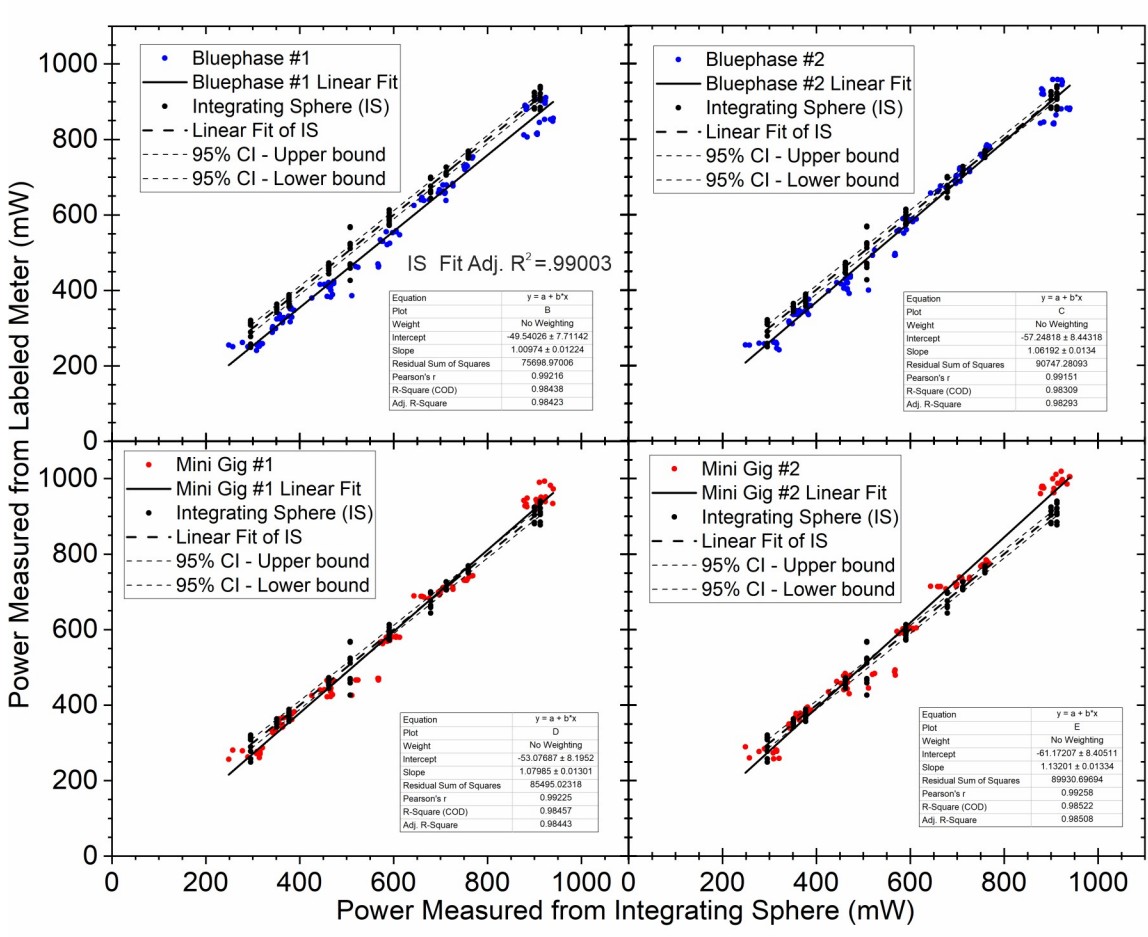

**Fig 9. Regression analysis and fitted line plots with 95% confidence and prediction intervals for mean power recordings of the 24 LCUs using the two Bluephase Meters II (units #1 and #2) and the two Mini Gig spectroradiometers (units #1 and #2).** The integrating sphere was the 'gold standard (GS)' predictor variable.

Although there were significant differences between the power values reported by the two hand-held radiometers used in this study and the third hypothesis was rejected on statistical grounds, Table 5 shows that overall, the mean differences between the power values recorded from the 24 LCUs using the two examples of the Bluephase Meter II were only 3.8 ± 1.5% and only 3.4 ± 1.3% for the more expensive Mini Gig spectroradiometer. Even the maximum differences in the power values from any the LCUs recorded by the two examples of each meter were at most 18% for the Bluephase Meter II and at most 15% for the Mini Gig spectroradiometer. All these large % differences were measured from light (C), which we classified as a 'budget LCU', whose standard deviations and coefficient of variation in the 'GS' power values were also large, suggesting that the output from this LCU was unstable. Since most of these differences between the meters illustrated in Figs 7 and 8 were well within the ±10% tolerance, we consider the small percentage differences between the meters to be acceptable for the dental office.

Based on our study, the Mini Gig radiometer was more precise and accurate than the Bluephase Meter II when reporting power. In view of the difference in cost and the fact that this

meter meets the the ISO/IEC 17025 standard [35, 36], this result is hardly suprising. When example #1 of each LCU was measured, six LCUs measured with Mini Gig were not significantly different from those measured using the 'GS' integrating sphere. When example #2 of each LCU was measured, eight LCUs measured with the Mini Gig radiometer were not significantly different from the 'GS' values (p>0.05). Furthermore, Fig 7 shows that when measuring example #1 of each LCU using the Mini Gig, the difference in the mean power values compared to the 'GS' value was less than 10% for all 24 measurements. Fig 8 shows that when measuring example #2 of each LCU using the Mini Gig, in 21 instances, the difference was 10% or less. When compared to the 'GS' values, the regression analyses showed that all four meters had $R^2$ values >0.98 (Fig 9). Therefore, the fourth hypothesis was accepted. This result is different from previous studies [28, 31] that reported the Bluephase Meter II was less accurate when measuring the output from some single peak LED LCUs. When the Bluephase Meter II manufacturer was contacted, the authors were informed that the meter had been upgraded, but the name had not been changed. From the slope equations reported in Fig 9, on average, the power values from the Bluephase Meter II #1 could be multiplied by 1.0097, the Bluephase Meter II #2 by 1.0619, the Mini Gig #1 by 1.0798 and the Mini Gig #2 by 1.1320 to produce the similar power values to those reported using the 'gold standard' system. However, as shown in Figs 7 and 8, both meters reported some power values that were slightly above and some below the 'GS' value. Consequently, a simple multiplication factor will not always produce an accurate power value, as it may be too high. Of note, the Mini Gig could accurately measure the power values of Pinkwave (a 2% difference at most), while Bluephase Meter II noticeably overestimated by ~50% the power output from the Pinkwave compared to those reported from the 'GS' method. The lack of accuracy when measuring the power values of Pinkwave with Bluephase Meter II is almost certainly because the manufacturer limits the measurement range of the Bluephase Meter II from 380 to 550 nm and emission spectrum from the Pinkwave extends far beyond this range. In contrast, the Mini Gig could accurately measure the Pinkwave (at most a 2% difference from the GS) because its measurement range is from 350 to 830 nm.

In a recent study, 50% of dentists used the same light exposure technique no matter what the shade or opacity of the RBC and 13% of American dentists surveyed used the same standard approach for all light-curing conditions [33]. Similar findings were reported in a survey of German dentists, where 50% of dental offices used the same 20-s exposure time for all restorations, and 27% of dental offices a 10-s exposure time for all their restorations [18]. This is concerning because this means that they are delivering very different amounts of energy and wavelengths of light to their restorations (Tables 3 and 4 and Figs 3–6), yet they and their patients both expect that their resins will be adequately photocured. This may be why the gingival margins of Class II RBC restorations are reported to be the most prevalent site for new caries formation [5, 13]. This region is furthest away from the light tip [38] and it requires a good LCU and a good light-curing technique to deliver an adequate amount of energy to photocure the RBC at the gingival margins of Class II RBC restorations [7, 39]. Otherwise, the resin here is likely to receive insufficient energy. The wide range in the power output from the LCUs helps to explain why direct resin-based composite (RBC) restorations placed in posterior teeth in dental offices have an unexpectedly high failure rate [13, 15, 16]. This has enormous financial implications as the use of dental amalgam is phased down [11, 23, 40]. Therefore, we propose that there is a need for more education about light-curing and LCUs. This education should include more hands-on practical instruction how to use the LCU, and that dentists should learn how to regularly monitor the output from their LCU using an accurate radiometer [25, 28, 31].

This study also shows that measuring the output from the LCU is not easy. Even the best laboratory-grade equipment used under ideal conditions is only accurate to ± 2 to 3% [36].

This equates to an acceptable range of ± 30 mW at 1000 mW. This would represent a range in the irradiance values of ± 78 mW/cm$^2$ if the light tip was 7 mm in diameter (area = 0.385 cm$^2$). Thus, the fact that the differences in the mean power values from the two examples of each brand of LCU when measured on the 'GS' system were less than 5.3% in 11 out of the 12 brands of LCUs is excellent. The apparent errors in the power measurements observed in our study may be due to variability in how the LCUs were manually held on the radiometer, or they may be due to the instability of the light output from the LCU itself and not the radiometer. Although Fig 3 shows that the power output from the 12 LCUs can be stable (flat), the larger standard deviations and coefficient of variations in the power values (Tables 3 and 4) support the results from previous studies that some 'budget lights' cannot maintain a stable initial light output [37, 41, 42].

## 4.1 Recommendations

Ideally, researchers should use a calibrated laboratory-grade integrating sphere and a spectroradiometer to measure and report the spectral radiant power, total power, and the energy received by the specimens in experiments that use dental LCUs and resins. However, we found that the Mini Gig meter was an acceptable alternative that was an easy-to-use, self-contained laboratory-grade portable spectroradiometer that exported calibrated spectral radiant power, power, and irradiance values. If the researcher cannot afford costly optical laboratory equipment to measure the output from the LCU, then the Mini Gig spectroradiometer should be used (contact Ultradent at +1 801 553 4351 for more information). The researcher will then be able to accurately report the power, irradiance, and energy delivered as well as the emission spectrum from the LCU. This information will help the researcher from coming to incorrect and possibly misleading conclusions.

However, due to the cost and the challenges posed when using and maintaining such laboratory-grade equipment, it would be unreasonable to ask clinicians to purchase such a costly laboratory-grade radiometer. We found that the Bluephase Meter II was easy to use, and although it could not report the emission spectrum, we found that it could accurately report the power from 11 out of the 12 brands of LCU tested. Knowing the power output from the LCU will help the clinician recognize that although two LCUs may deliver the same irradiance, the actual power output may easily be 50% less from the LCU that has a smaller tip diameter. Furthermore, when the power is multiplied by the exposure time (s), the energy (in Joules) from the LCU can be calculated.

## 4.2 Final comments and possible directions for further development

This study highlights that measuring the output from dental LCUs is not as easy as it seems. Even the best laboratory-grade equipment will only be accurate to ± 20 to 30 mW at 1000 mW [36]. Thus, while there may be statistical differences between the power values reported by the meters in this study, these differences were often well within the tolerances of the measuring equipment, and they were also within the meter manufacturer's tolerances.

A limitation of this study is that the Bluephase Meter II and the Mini Gig radiometers were brand new, and how long they will remain calibrated is unknown. Also, only two examples of each meter and radiometer were measured. The LCUs were not new, but they were undamaged, in good working order, they were recharged after every five exposures. Thus, they represented what may be found in many dental offices. Finally, the LCUs were only tested on their standard power output setting. Despite these study limitations, both radiometers could accurately measure the power from a wide range of contemporary LCUs that had large differences in their power output and emission spectra.

## 5. Conclusions

Within the limitations of this study, we concluded that there was a large and significant (p<0.001) difference in the power values (mW) and the emission spectra emitted by 12 brands of LCUs. Five of the LCUs emitted less than 60% of the power emitted by the most powerful LCU. Except for the Dental Spark LCU, where there was a 15.1% difference, the differences between the power values from two units of the same brand of LCU were less than 5.3% when measured using the 'GS' integrating sphere system. The overall mean ± S.D. difference between the power values from the same LCU recorded by the two examples of the Bluephase Meter II was 3.8 ± 1.5% and 3.4 ± 1.3% for the more expensive Mini Gig spectroradiometer. Finally, we found that provided that the meters were not attempting to measure wavelengths beyond their design specifications, the two brands of dental radiometers tested were accurate and precise.

## Supporting information

**S1 Data.**
(ZIP)

## Author Contributions

**Conceptualization:** Richard B. Price.

**Data curation:** Cristiane Maucoski.

**Formal analysis:** Cristiane Maucoski, Richard B. Price, Cesar A. Arrais.

**Investigation:** Cristiane Maucoski, Richard B. Price, Braden Sullivan.

**Methodology:** Cristiane Maucoski, Richard B. Price.

**Resources:** Richard B. Price.

**Supervision:** Richard B. Price, Cesar A. Arrais.

**Validation:** Richard B. Price.

**Visualization:** Richard B. Price.

**Writing – original draft:** Cristiane Maucoski, Richard B. Price, Cesar A. Arrais.

**Writing – review & editing:** Cristiane Maucoski, Richard B. Price, Cesar A. Arrais.

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
