## [Decision Letter · Decision Letter 0]

8 Feb 2022

PONE-D-21-40118Light Output from 12 Brands of Contemporary LED Light-Curing Units Measured Using 2 Brands of RadiometersPLOS ONE

Dear Dr. Price,

Thank you for submitting your manuscript to PLOS ONE. After careful consideration, we feel that it has merit but does not fully meet PLOS ONE’s publication criteria as it currently stands. Therefore, we invite you to submit a revised version of the manuscript that addresses the points raised during the review process. Please submit your revised manuscript by Mar 25 2022 11:59PM. If you will need more time than this to complete your revisions, please reply to this message or contact the journal office at plosone@plos.org. Please include the following items when submitting your revised manuscript:A rebuttal letter that responds to each point raised by the academic editor and reviewer(s). You should upload this letter as a separate file labeled 'Response to Reviewers'.A marked-up copy of your manuscript that highlights changes made to the original version. You should upload this as a separate file labeled 'Revised Manuscript with Track Changes'.An unmarked version of your revised paper without tracked changes. You should upload this as a separate file labeled 'Manuscript'.If applicable, we recommend that you deposit your laboratory protocols in protocols.io to enhance the reproducibility of your results. Protocols.io assigns your protocol its own identifier (DOI) so that it can be cited independently in the future. For instructions see: https://journals.plos.org/plosone/s/submission-guidelines#loc-laboratory-protocols. Additionally, PLOS ONE offers an option for publishing peer-reviewed Lab Protocol articles, which describe protocols hosted on protocols.io. Read more information on sharing protocols at https://plos.org/protocols?utm_medium=editorial-email&utm_source=authorletters&utm_campaign=protocols.

We look forward to receiving your revised manuscript.

Kind regards,

Zhiyuan Zhu

Academic Editor

PLOS ONE

Journal Requirements:

[The authors wish to thank the manufacturers for the loan of their Bluephase Meter II and the Mini Gig radiometers for this study. This study was supported by a travel grant provided by MITACS award IT26826 for C. Maucoski and an internal research fund grant from the Faculty of Dentistry, Dalhousie University. The study was also supported by grant provided by Coordination for the Improvement of Higher Education Personnel (CAPES grant # 88881.622852/2021-01).]

[The authors received no specific funding for this work.

Cristiane Maucoski is a graduate student from Brazil. She received a travel grant provided by a MITACS award IT26826 for C. Maucoski and an internal research fund grant from the Faculty of Dentistry, Dalhousie University. The study was also supported by grant provided by Coordination for the Improvement of Higher Education Personnel (CAPES grant # 88881.622852/2021-01).  

The funders had no role in study design, data collection and analysis, decision to publish, or preparation of the manuscript.]

5. We note you have included a table to which you do not refer in the text of your manuscript. Please ensure that you refer to Table 2 in your text; if accepted, production will need this reference to link the reader to the Table.

Reviewers' comments:

Reviewer's Responses to Questions

**Comments to the Author**

1. Is the manuscript technically sound, and do the data support the conclusions?

Reviewer #1: Yes

Reviewer #2: Yes

2. Has the statistical analysis been performed appropriately and rigorously? 

Reviewer #1: Yes

Reviewer #2: Yes

3. Have the authors made all data underlying the findings in their manuscript fully available?

Reviewer #1: Yes

Reviewer #2: Yes

4. Is the manuscript presented in an intelligible fashion and written in standard English?

Reviewer #1: Yes

Reviewer #2: Yes

5. Review Comments to the Author

Reviewer #1: ***Please download the attachment and ignore the following words.***

1.

In citation,

29.Worthington HV, Khangura S, Seal K, Mierzwinski-Urban M, Veitz-Keenan A, Sahrmann P, et al. Direct composite resin fillings versus amalgam fillings for permanent posterior teeth. Cochrane Database Syst Rev. 2021;8:CD005620. Epub 20210813. doi: 10.1002/14651858.CD005620.pub3. PubMed PMID: 34387873; PubMed Central PMCID: PMCPMC8407050.

There’s your sentences:

M E T H O D S

Criteria for considering studies for this review

Types of studies

For the outcome of efficacy, we included randomised controlled trials (RCTs) comparing dental composite resin with dental amalgam restorations in permanent posterior teeth (dating back to 1946). We excluded studies that had less than a three-year followup period.

Direct composite resin fillings versus amalgam fillings for permanent posterior teeth . This article written in 2021,but the cited literature’s date is too old, which I think is inappropriate, and I suggest you cite another references.

2.

As the LCU usage time increasing, the attenuation of the radiant output should also be taken into account.

3.

The effect of voltage on light intensity should be considered. For example, a 100% battery will produce the highest voltage intensity. A battery with a 10% charge will produce a smaller voltage.

In clinical, it is impossible for us to keep the LCU at 100% power all the time.

4.

The following sentence might be a sick sentence:  For LCU units #1, the following lights showed percentage differences beyond 10%: Dental Spark, SDI Radii Plus and Woodpecker LED.B. For LCU units #2, these lights showed percentage differences beyond 10%: Dental Spark, SDI Radii Plus, Woodpecker LED. B and Woodpecker LED.D. 

5.

Did you consider the effect of temperature on the test during the experiment?

6.

The photoinitiator system is composed of photoinitiator and co-initiator, which plays a decisive role in the photocuring rate of the material. The most commonly used photoinitiator is camphoroquinone (CQ, camphoroquinone), the addition amount is usually 0.05%-1% (mass fraction). Camphorquinone absorbs blue light in the range of 400-500 nm, with a maximum absorption peak at 470 nm.

The entire article focuses on the radiated power of LCUs. I think the peaks of each spectrum are the ones to focus on.

More attention should be paid to clinical practice. What we want is the curing speed and depth, not the rigid value of radiation power.

Reviewer #2: The authors measured the emitted power of 12 brands of contemporary light-curing units using two different brands of dental radiometers (Bluephase Meter II and Mini Gig) and compared to a "GOLD STANDARD" fiberoptic spectroradiometer (Ocean Insight) in order to evaluate the accuracy and precision of measured power value of the dental radiometers. The authors concluded that the radiometers were not attempting to measure wavelengths beyond their design specifications, the two brands of dental radiometers tested were accurate and precise. These findings will be of interest to dental practitioners, as well as researchers in the field.

I have only one concern;

1. Page 9, 2nd paragraph of Introduction: "If these light-cured resins are not sufficiently photocured

they may not reach their intended properties[13, 14], the economic impact of millions of

inadequately photopolymerized RBC restorations failing earlier than expected and requiring

retreatment is enormous." Since this is the first time you use "RBC", the authors should not use abbreviation, but describe "resin-based composite (RBC)".

6. PLOS authors have the option to publish the peer review history of their article (what does this mean?). If published, this will include your full peer review and any attached files.

Reviewer #1: No

Reviewer #2: No

---

## [Author Response · Author response to Decision Letter 0]

28 Feb 2022

February 24, 2022

PLOSOne 

Dr. Zhiyuan Zhu

Southwest University

CHINA 

Dear Dr. Zhu:

RE: PONE-D-21-40118 ‘Light Output from 12 Brands of Contemporary LED Light Curing Units Measured Using 2 Brands of Radiometers’ 

Thank you for giving us the opportunity to submit a revision to our manuscript to PLOS One.

The purpose of the article is to alert your readers to five important points:

1. Using a malfunctioning or an inadequate dental curing light has an economic impact estimated to cost billions of dollars annually.

2. Dental curing lights should be tested regularly.

3. The Bluephase Meter II can accurately measure the power from most dental curing lights and should be used by dentists

4. If the researcher cannot afford and cannot access optics laboratory equipment, the Mini Gig radiometer should be used. This will provide the power, spectral radiant power and the emission spectrum.

We have uploaded a clean and a tracked version, but to answer your reviewer’s specific comments:

1. We apologize for any confusion. Dr. Cristiane Maucoski is from Brazil. She received a Globalink grant of $6,000 from MITACS to come to Canada. The money was not specific to the research presented in the manuscript, but without the MITACS Globalink support she could not have come to Canada. As such, we believe that the following ‘Funding Information’ and ‘Financial Disclosure’ statements are now correct. The authors wish to thank the manufacturers for the loan of their Bluephase Meter II and the Mini Gig radiometers for this study. Dr. Maucoski’s visit to Canada was supported by grants from the Brazilian Coordination for the Improvement of Higher Education Personnel (CAPES grant # 88881.622852/2021-01) and by a MITACS Globalink award IT26826. This study was also supported by an internal research grant from the Faculty of Dentistry, Dalhousie University, Canada.

2. To address the concerns that the 2021 Cochrane review was too old, we have added additional references and reworded the paragraph. However, the meaning is still the same, RBCs do not last as long as they should. Why?

This may explain why a Cochrane review published in 2021 reported that direct resin-based composite (RBC) restorations placed in posterior teeth have almost double the failure rate of amalgam restorations[31]. Several previous publications have reported higher failure rates for RBCs [18, 32] and that the median longevity of posterior RBCs is less than 7 years [33, 34].

3. To address the concern that as the LCU usage time increasing, the attenuation of the radiant output should also be taken into account. And The effect of voltage on light intensity should be considered. For example, a 100% battery will produce the highest voltage intensity. A battery with a 10% charge will produce a smaller voltage.

We addressed this when we wrote ‘The LCUs were recharged after every 5 exposures to ensure that their batteries were always adequately charged.’

4. To address the concern that For LCU units #1, the following lights showed percentage differences beyond 10%: Dental Spark, SDI Radii Plus and Woodpecker LED.B. For LCU units #2, these lights showed percentage differences beyond 10%: Dental Spark, SDI Radii Plus, Woodpecker LED. B and Woodpecker LED.D, we revised the sentence to read: Examples #1 and 2 of the Dental Spark, SDI Radii Plus and Woodpecker LED.B both showed percentage differences in the power output that were beyond 10%. Only example #2 of the Woodpecker LED.D had percentage differences in the power output that were beyond 10%. We reworded the section to read:

3.3 Comparison between LCUs

Figures 7 and 8 illustrate the percentage differences of each radiometer compared to the ‘GS’ measurement for each LCU. Of note, both meters reported power values that were above and below the ‘GS’ value. Examples #1 and 2 of the Dental Spark, SDI Radii Plus and Woodpecker LED.B both showed differences in the power output that were more than 10% different from the ‘GS’. Example #2 of the Dental Spark, SDI Radii Plus, Woodpecker LED.B and the Woodpecker LED.D had percentage differences in the power output that were greater than 10% different from the ‘GS’. Apart from the SDI Radii Plus, we considered these to be 'budget lights’. Figures 7 and 8 show that the highest percentage difference was for the Pinkwave when using the Bluephase Meter II, and these values were obviously incorrect. For the Mini Gig meter #1, the greatest difference from the 'GS’ result was 9%. For the Mini Gig meter #2, the greatest difference from the 'GS' was 15%. Both of these values were from the budget light Dental Spark LCU (c), whose standard deviations in the ‘GS’ power values were also large (±25 and 30 mW). 

5. To address the concern, Did you consider the effect of temperature on the test during the experiment? We added:

The LCUs and meters were used in random order and at room temperature (20°C±1) to mimic the condition in a dental office.

6. To address the concern, More attention should be paid to clinical practice. What we want is the curing speed and depth, not the rigid value of radiation power. We believe that this reviewer has missed the point. Dentists and researchers need a reliable and accurate instrument to measure the light output from their light. This article tells the dentist and the researcher what to buy. The dentist ‘should be confident purchasing the Bluephase Meter II because it could accurately report the power from 11 out of the 12 brands of LCU tested.’ The researcher should purchase the Mini Gig. We have significantly edited the manuscript and hoe that this meets the reviewer’s concerns. Following our recommendations will enable dentists to make better treatment decisions that are based on the power (Watts) and amount of energy (Joules) that can be delivered to the RBC from the LCU.

7. We thank the reviewer for pointing out that we had not described what RBC meant before using the term. This error has been corrected. 

8. We thank the reviewer for pointing out that we had not described Table 2 in the text. This has been corrected.

9. If our manuscript is accepted for publication, we shall upload our laboratory protocols to protocols.io to enhance the reproducibility of our results. 

Sincerely yours,

R.B. Price, BDS, DDS, MS, FRCD(C), PhD

Professor of Prosthodontics

---

## [Decision Letter · Decision Letter 1]

21 Mar 2022

PONE-D-21-40118R1Light Output from 12 Brands of Contemporary LED Light-Curing Units Measured Using 2 Brands of RadiometersPLOS ONE

Dear Dr. Price,

Thank you for submitting your manuscript to PLOS ONE. After careful consideration, we feel that it has merit but does not fully meet PLOS ONE’s publication criteria as it currently stands. Therefore, we invite you to submit a revised version of the manuscript that addresses the points raised during the review process.

We look forward to receiving your revised manuscript.

Kind regards,

Zhiyuan Zhu

Academic Editor

PLOS ONE

Journal Requirements:

Reviewers' comments:

Reviewer's Responses to Questions

**Comments to the Author**

1. If the authors have adequately addressed your comments raised in a previous round of review and you feel that this manuscript is now acceptable for publication, you may indicate that here to bypass the “Comments to the Author” section, enter your conflict of interest statement in the “Confidential to Editor” section, and submit your "Accept" recommendation.

Reviewer #1: (No Response)

Reviewer #2: All comments have been addressed

2. Is the manuscript technically sound, and do the data support the conclusions?

Reviewer #1: Yes

Reviewer #2: Yes

3. Has the statistical analysis been performed appropriately and rigorously? 

Reviewer #1: Yes

Reviewer #2: Yes

4. Have the authors made all data underlying the findings in their manuscript fully available?

Reviewer #1: Yes

Reviewer #2: Yes

5. Is the manuscript presented in an intelligible fashion and written in standard English?

Reviewer #1: Yes

Reviewer #2: Yes

6. Review Comments to the Author

Reviewer #1: please download the attachment

please download the attachment

please download the attachment

please download the attachment

Reviewer #2: (No Response)

7. PLOS authors have the option to publish the peer review history of their article (what does this mean?). If published, this will include your full peer review and any attached files.

Reviewer #1: No

Reviewer #2: No

---

## [Author Response · Author response to Decision Letter 1]

4 Apr 2022

April 4, 2022

PLOSOne 

Dr. Zhiyuan Zhu

Southwest University

CHINA 

Dear Dr. Zhu:

Revisions to:

PONE-D-21-40118R2

Light Output from 12 Brands of Contemporary LED Light-Curing Units Measured Using 2 Brands of Radiometers

Enclosed is our revised manuscript that we would like to resubmit to PLOS One. 

Comment #1:

Have you ever pressed a button manually？If you press a button with your finger, how do you make sure the timing of each button press is very accurate? 

Comment #2:

Do you have the spectrogram similar to this picture below? If the answer is yes, please add into the article.

Answers:

1. We have addressed the concern about the 10-s. The exact timing was not relevant to the study, but we have addressed this concern in the text.

2. As requested, we have added a colour spectrum to Figure 4 and added it as a striking image.

The authors received no specific funding for this work, but the authors wish to thank the manufacturers for the loan of their Bluephase Meter II and the Mini Gig radiometers used in this study. 

Dr. Cristiane Maucoski is a visiting graduate student from Brazil. She received a travel grant provided by a MITACS award IT26826 and an internal research fund grant from the Faculty of Dentistry, Dalhousie University. Her visit to Canada was also supported by grant provided by Coordination for the Improvement of Higher Education Personnel (CAPES grant # 88881.622852/2021-01). Neither the manufacturers of the radiometers nor the funders of Dr. Maucoski’s visit to Canada had any role in the study design, data collection and analysis, decision to publish, or preparation of the manuscript.

Sincerely yours,

R.B. Price, BDS, DDS, MS, FRCD(C), PhD

Professor of Prosthodontics

---

## [Editor Report · Decision Letter 2]

7 Apr 2022

Light Output from 12 Brands of Contemporary LED Light-Curing Units Measured Using 2 Brands of Radiometers

PONE-D-21-40118R2

Dear Dr. Price,

We’re pleased to inform you that your manuscript has been judged scientifically suitable for publication and will be formally accepted for publication once it meets all outstanding technical requirements.

Kind regards,

Zhiyuan Zhu

Academic Editor

PLOS ONE
---

## [Editor Report · Acceptance letter]

30 Jun 2022

PONE-D-21-40118R2 

Power output from 12 brands of contemporary LED light-curing units measured using 2 brands of radiometers 

Dear Dr. Price:

I'm pleased to inform you that your manuscript has been deemed suitable for publication in PLOS ONE. Congratulations! Your manuscript is now with our production department. 

Kind regards, 

on behalf of

Prof. Zhiyuan Zhu 

Academic Editor

PLOS ONE